



**Ice core evidence for decoupling between mid-latitude atmospheric water cycle and Greenland**
**temperature during the last deglaciation**
Amaëlle Landais[1,*], Emilie Capron[2,3], Valérie Masson-Delmotte[1], Samuel Toucanne[4], Rachael
Rhodes[5], Trevor Popp[2], Bo Vinther[2], Bénédicte Minster[1], Frédéric Prié[1]
[1] Laboratoire des Sciences du Climat et de l'Environnement, IPSL, UMR 8212, CEA-CNRS-UVSQ-UPS,
Gif sur Yvette, France
[2] Centre for Ice and Climate, Niels Bohr Institute, University of Copenhagen, Juliane Maries Vej 30,
DK-2900, Copenhagen, Denmark;
[3] British Antarctic Survey, High Cross, Madingley Road, Cambridge, CB3 0ET, UK
[4] IFREMER, Laboratoire Géophysique et enregistrement Sédimentaire, CS 10070, 29280 Plouzané,
France
[5] Department of Earth Sciences, University of Cambridge, Downing Street, Cambridge, CB2 3EQ
UK
* Corresponding author: phone : +33 (0)169084672, email : amaelle.landais@lsce.ipsl.fr, ORCID :

16 0000-0002-5620-5465

Keywords: Deglaciation, Water isotopes, $^{17}O$-excess, d-excess, Heinrich event, Mystery interval



## Abstract

The last deglaciation represents the most recent example of natural global warming associated with large-scale climate changes. In addition to the long-term global temperature increase, the last deglaciation onset is punctuated by a sequence of abrupt changes in the Northern Hemisphere. Such interplay between orbital- and millennial-scale variability is widely documented in paleoclimatic records but the underlying mechanisms are not fully understood. Limitations arise from the difficulty in constraining the sequence of events between external forcing, high- and low- latitude climate and environmental changes.

Greenland ice cores provide sub-decadal-scale records across the last deglaciation and contain fingerprints of climate variations occurring in different regions of the Northern Hemisphere. Here, we combine new ice d-excess and $^{17}$O-excess records, tracing changes in the mid-latitudes, with ice $\delta^{18}$O records of polar climate. Within Heinrich Stadial 1, we demonstrate a decoupling between climatic conditions in Greenland and those of the lower latitudes. While Greenland temperature remains mostly stable from 17.5 to 14.7 ka, significant change in the mid latitudes of northern Atlantic takes place at ~16.2 ka, associated with warmer and wetter conditions of Greenland moisture sources. We show that this climate modification is coincident with abrupt changes in atmospheric $CO_2$ and $CH_4$ concentrations recorded in an Antarctic ice core. Our coherent ice core chronological framework and comparison with other paleoclimate records suggests a mechanism involving two-step freshwater fluxes in the North Atlantic associated with a southward shift of the intertropical convergence zone.




### **Introduction**

The last deglaciation (~19 thousand to 11 thousand years before present, ka) is the most recent major
reorganization of global climate and is thus extensively documented by proxy records from natural
climate archives. The wealth of high-resolution records from well-dated archives and data synthesis
obtained over the past decades show two modes of climate variability during this period (e.g. Denton
et al., 2010, Clark et al., 2012). The first is a long-term increase in global surface temperature and
atmospheric $CO_2$ concentration between 18 and 11 ka. Superimposed on this is a sequence of
centennial-scale transitions between three quasi-stable intervals documented in Northern
Hemisphere temperature, namely (i) the Heinrich Stadial 1 (~17.5-14.7 ka), that encompasses the
massive rafting episode known as Heinrich event 1 (from ~16 ka); (ii) the Bølling-Allerød warming phase
(~14.7 to 12.9 ka) and (iii) the Younger Dryas cold phase (~12.9 to 11.7 ka). This three-step sequence
coincides with rapid variations in the Atlantic Meridional Oceanic Circulation (AMOC) (Mc Manus et
al., 2004), with evidence for a weak meridional overturning in the North Atlantic during the cold period
encompassing Heinrich Stadial 1 and the Younger Dryas.
Our understanding of the mechanisms at play during these North Atlantic cold phases remains limited.
First, recent studies challenge the earlier attribution of the AMOC slowdown during Heinrich Stadial 1
to the impact of the Iceberg Rafted Debris (IRD) from the Laurentide ice sheet through Hudson Strait
(Alvarez-Solas et al., 2011). In particular, meltwater releases from the European ice sheet occurring as
early as 19 or 20 ka may have played an important role in this AMOC slowdown (Toucanne et al., 2010;
Stanford et al., 2011; Hodell et al., 2017).
Second, major global reorganizations of the hydrological cycle have been demonstrated during
Heinrich Stadial 1. They can be separated in two phases. In North America, a first time interval
characterized by low lake levels (referred to as the "big dry", 17.5 to 16.1 ka) was followed by a second
time interval with high lake levels (referred to as the "big wet", 16.1 to 14.7 ka) (Broecker et al., 2012),



both apparently occurring during a stable cold phase in Greenland temperature. The second phase of
Heinrich Stadial 1 is also associated with a weak East Asian monsoon interval (Zhang et al., 2014),
understood to reflect a southward shift of the Inter-tropical Convergence Zone (ITCZ). While there is
growing evidence for large-scale reorganizations of climate and low- to mid- latitude atmospheric
water cycle within Heinrich Stadial 1, the exact sequence of events is not known with sufficient
accuracy to understand the links between changes in North Atlantic climate, AMOC, and the lower
latitude water cycle.
Linking changes in the high latitudes of the North Atlantic and the mid- to low- latitudes requires
precise absolute chronologies such as those obtained from annual layer counting of Greenland ice (e.g.
Andersen et al., 2006) or U/Th dating of speleothems (e.g. Zhang et al., 2014). Unfortunately, absolute
dating uncertainties increase above a hundred years during the last deglaciation, precluding a direct
comparison of proxy records at the centennial scale. In this study, we circumvent this difficulty by using
a diverse range of proxy records measured on Greenland ice cores that represent both Greenland
temperature and mid-latitude moisture source conditions.
**Analytical method**
Here, we present new water isotope records ($\delta^{18}$O, d-excess, $^{17}$O-excess) from the NGRIP ice core
(NGRIP et al., 2004), reported on the annual-layer counted Greenland Ice Core Chronology 2005
(hereafter GICC05, Rasmussen et al., 2006; Svensson et al., 2008), and associated with relatively small
absolute uncertainties over the last deglaciation (maximum counting 1σ error of 100-200 yr). Other
Greenland and Antarctic ice cores have been aligned on the GICC05 chronology, with a maximum
relative dating uncertainty of 400 years over the last deglaciation (Rasmussen et al., 2008; Bazin et al.n
2013; Veres et al., 2013).
The new NGRIP $\delta^{18}$O and $\delta$D dataset was obtained at Laboratoire des Sciences du Climat et de
l'Environnement (LSCE) using a laser cavity ring-down spectroscopy (CRDS) analyzer PICARRO. The
accuracy for $\delta^{18}$O and $\delta$D measurements displayed here is about 0.1‰ and 1‰ respectively. This new





dataset completes the NGRIP high-resolution isotopic dataset published over the time period 11.5 to
14.7 ka with $\delta^{18}O$ and $\delta D$ measured respectively at the University of Copenhagen and at the Institute
of Arctic and Alpine Research (INSTAAR) Stable Isotope Lab (SIL) (University of Colorado), respectively.
$\delta^{18}O$ analyses were performed at the Niels Bohr Institute (University of Copenhagen) using a $CO_2$
equilibration technique (Epstein et al., 1953) with an analytical precision of 0.07‰. $\delta D$ measurements
at INSTAAR were made via an automated uranium reduction system coupled to a VG SIRA II dual inlet
mass spectrometer (Vaughn et al., 1998). Analytical precision for $\delta D$ is ±0.5‰ or better. Both series
show similar $\delta^{18}O$ values, in agreement with the reference $\delta^{18}O$ series for NGRIP over the last climatic
cycle (NGRIP community members, 2004) within error bars. However, while both LSCE and INSTAAR
SIL d-excess series display the same 3.5‰ decrease over the onset of Bølling-Allerød, the mean d-
excess level differs by 2.5‰ between the two timeseries. Despite several home standard
intercalibrations between the two laboratories, this difference remains unexplained and prevents any
further discussion on the absolute NGRIP d-excess levels. The new and published NGRIP d-excess
dataset are combined after a shift of the INSTAR SIL d-excess series by -2.5‰.
In order to perform $^{17}O$-excess measurements on water samples at LSCE, water reacts with $CoF_3$ to
produce oxygen whose triple isotopic composition is measured by dual inlet against a reference $O_2$ gas
resulting in a mean uncertainty of 5 ppm (1 $\sigma$) for the $^{17}O$-excess measurements (Barkan and Luz,
2005). Every day, at least one home standard is run with the batch of samples to check the stability of
the fluorination line and mass spectrometer and a series of water home standards whose $\delta^{18}O$
encompasses the SMOW – SLAP scale is run every month enabling to calibrate the $\delta^{18}O$ and $^{17}O$-excess
values (Schoenemann et al., 2013).
**Results**
Ice core $\delta^{18}O$ (NGRIP community members, 2004) is a qualitative proxy for local surface temperature.
Comparisons between ice core $\delta^{18}O$ data and paleotemperature estimates from borehole temperature
profile inversion and abrupt temperature changes inferred from isotopic measurements on trapped



air showed that the $\delta^{18}O$-temperature relationship at NGRIP varies from 0.3 to 0.5 ‰.°C$^{-1}$ during
glacial-interglacial periods (Buizert et al., 2014; Dahl-Jensen et al., 1998; Kindler et al., 2014). In
addition to $\delta^{18}O$ records already available (NGRIP community members, 2004), we provide here new
d-excess data from NGRIP during the last deglaciation. The second-order parameter d-excess ($\delta D$-
$8x\delta^{18}O$) (Dansgaard, 1964) is used in Greenland ice cores to track past changes in evaporation
conditions or shifts in moisture sources (Johnsen et al., 1989; Masson-Delmotte et al., 2005a).
Evaporation conditions affect the initial vapor d-excess through the impact of surface humidity and
sea surface temperature on kinetic fractionation (Jouzel et al., 1982). Recent vapour monitoring and
modelling studies show that the d-excess signal of the moisture source can be preserved in polar
vapour and precipitation after transportation towards polar regions (Bonne et al., 2015; Pfahl and
Sodemann, 2014). This signal can however be altered during distillation due to the sensitivity of
equilibrium fractionation coefficients to temperature, leading to alternative definitions using
logarithm formulations for Antarctic ice cores (Uemura et al., 2012; Markle et al., 2016). Finally,
changes in $\delta^{18}O_{sea\ water}$ also influence $\delta^{18}O$ and d-excess in polar precipitation. Summarizing, d-excess in
Greenland ice core is a complex tracer: interpreting its past variations in terms of changes in
evaporation conditions (sea surface temperature or humidity) requires deconvolution of the effects of
glacial-interglacial changes in $\delta^{18}O_{sea\ water}$ and condensation temperature.
Our dataset also encompasses new $^{17}O$-excess data from NGRIP. Defined as $\ln(\delta^{17}O+1)$-
$0.528*\ln(\delta^{18}O+1)$), $^{17}O$-excess provides complementary information to d-excess (Landais et al., 2008;
Landais et al., 2012). At evaporation, d-excess and $^{17}O$-excess are both primarily influenced by the
balance between kinetic and equilibrium fractionation, itself driven by relative humidity at the sea
surface. During transport, while d-excess is influenced by distillation effects during atmospheric
cooling, $^{17}O$-excess is largely insensitive to this effect, except at very low temperatures in Antarctica
(Winkler et al., 2012). Conversely, $^{17}O$-excess is affected by recycling or mixing of air masses along the
transport path from low to high latitudes (Risi et al., 2010), and by the range over which supersaturated
conditions occur, itself affected for instance by changes in sea-ice extent or temperature along the



transport path (Schoenemann et al., 2014). Because of its logarithmic definition, $^{17}$O-excess is not
sensitive to changes in $\delta^{18}O_{sea\ water}$ given that the $^{17}$O-excess of global sea water remains constant with
time.
Our 1518 new measurements of $\delta^{18}O$ and d-excess on the NGRIP ice core cover the time period 14.5
to 60 ka (Figure 1) and we present 454 duplicate measurements of $^{17}$O-excess over the time period
ranging from 9.6 to 20 ka (Figure 2) (see methods for details). As previously reported for the central
Greenland GRIP ice core (Masson-Delmotte et al., 2005b; Jouzel et al., 2005), the NGRIP $\delta^{18}O$ and d-
excess records exhibit a systematic anti-correlation during the abrupt Dansgaard-Oeschger (DO) events
of the last glacial period and last deglaciation (Bølling-Allerød and Younger Dryas), with d-excess being
higher during cool Greenland Stadials and lower during warm Greenland Interstadials.
The origin of moisture may be different at GRIP and NGRIP. While both sites are expected to receive
most of their moisture from the North Atlantic (30°N to 55°N, Landais et al., 2012) with modulation
partly linked to sea ice extent (Rhines et al., 2014), the northwestern NGRIP site may also receive
moisture from North Pacific (Langen and Vinther, 2009). Nevertheless, the two sites depict similar
amplitudes of d-excess variations across DO events (Figure 1). We note that this contrasts with a
slightly lower amplitude (typically by 1‰) of abrupt $\delta^{18}O$ changes at NGRIP compared to GRIP.
The fact that d-excess increases (by 3.5 ± 1 ‰) when $\delta^{18}O$ decreases (by 4 ± 1 ‰) during Greenland
stadials relative to interstadials may at least partly reflects the influence of local temperature changes
on d-excess, challenging a simple interpretation in terms of changes in source conditions. We note one
exception, the Heinrich Stadial 1 cold phase preceding the onset of the Bølling-Allerød at 14.7 ka. In
this case, $\delta^{18}O$ remains almost stable from 17.5 to 14.7 ka on the three Greenland ice cores NGRIP,
GRIP and GISP2 displayed on Figure 2. Over this period, $\delta^{18}O$ variations are smaller than 1 ‰, i.e. less
than one fourth of the average amplitude in $\delta^{18}O$ changes across DO events, suggesting no large
temperature change in Greenland during this period. The link between flat $\delta^{18}O$ and minimal
temperature variability can be challenged since a mean temperature signal can be masked by a change



in seasonality of moisture source origin on the $\delta^{18}$O record (Boyle et al., 1994; Krinner et al., 1997).
However, our assumption of stable temperature is supported by constant $\delta^{15}$N of $N_2$ values in the GISP2
and NGRIP ice cores (Buizert et al., 2014), $\delta^{15}$N of $N_2$ being an alternative paleothermometry tool in ice
core that is not affected by processes within the water cycle (Severinghaus and Brook, 1999). In
contrast to this almost stable $\delta^{18}$O signal, d-excess depicts an average 2.2 ‰ increase at 16.1 ka (more
than 60% of the average amplitude during DO events) with a larger amplitude at GRIP (2.7 ‰) than at
NGRIP (1.7 ‰) (Figure 2). In this case, the increase in d-excess cannot be explained by any Greenland
temperature change, and therefore demonstrates a decoupling between cold and stable Greenland
temperatures and changing climatic conditions at lower latitudes during Heinrich Stadial 1 (see also
SOM).
While the $^{17}$O-excess level is similar at the Last Glacial Maximum (i.e. before 19 ka on Figure 2) and the
Early Holocene (40 ppm), it also shows significant variations during the last deglaciation. Most of these
variations co-vary with those of $\delta^{18}$O such as the four main oscillations during the Bølling-Allerød and
the onset and end of the Younger Dryas. They can be interpreted as parallel variations in the Greenland
temperature and lower latitude climate with a possible contribution of local temperature through
kinetic effects. Again, a major difference occurs during Heinrich Stadial 1. While the $\delta^{18}$O record is
relatively stable, $^{17}$O-excess exhibits a decreasing trend (strongest between 17.5 and 16.1 ka) before a
minimum level is reached between 16.1 to 14.7 ka. We therefore observe a clear and synchronous
signal in both d-excess and $^{17}$O-excess dated around 16.2 ka from statistical analysis (cf. section
statistical analyses in SOM). These $^{17}$O-excess and d-excess transitions at 16.2 ka do not have any clear
counterpart in $\delta^{18}$O (cf section correlation in SOM) and no temperature variation at that time was
recorded in the $\delta^{15}$N of $N_2$ record. We interpret these patterns as illustrating a reorganization of
climatic conditions and/or water cycle at latitudes south of Greenland. A similar shift in $^{17}$O-excess has
already been observed during Heinrich Stadial 4 in the NEEM ice core, while the $\delta^{18}$O record exhibits a
constant low level (Guillevic et al., 2014). This pattern was also attributed to a change in the water
cycle and/or climate at lower latitudes.



**Discussion**
The Greenland water stable isotope records demonstrate a change in the water cycle and/or climate
at lower latitudes at 16.2 ka when Greenland conditions were relatively stable and cold. This change
at low latitudes is confirmed by the high resolution atmospheric $CH_4$ concentration record from the
WAIS Divide ice core (Rhodes et al., 2015), presented on the same timescale (Figure 2). At 16.2 ka, the
$CH_4$ record indeed exhibits a 30 ppbv peak understood to reflect more $CH_4$ production in Southern
Hemisphere wetlands, driven by wetter conditions due to a southward shift of the tropical rainbelts
associated with the ITCZ (Rhodes et al., 2915). The parallel increase of atmospheric $CO_2$ concentration
by 10 ppm in ~100 years (Marcott et al., 2013) is understood to result from increased terrestrial carbon
fluxes or enhanced air-sea gas exchange in the Southern Ocean (Bauska et al., 2014). We also highlight
an unusual characteristic of the bipolar seesaw pattern in Antarctic ice core $\delta^{18}O$ records at 16.2 ka. As
observed during all Greenland Stadials of the last glacial period, Antarctic $\delta^{18}O$ also increases during
Heinrich Stadial 1 (e.g. EPICA community members, 2006), through the warming phase of Antarctic
Isotopic Maximum 1. The EPICA Dronning Maud Land (EDML) ice core, drilled in the Atlantic sector of
Antarctica, shows an associated two step $\delta^{18}O$ increase. The first step, marked by a strong increasing
trend, is followed by a change in slope at 16.2 ka. The second step is characterized by a slower
increasing trend from 16.2 to 14.7 ka (EPICA community members, 2006; Stenni et al., 2011) (Figure
2). The EDML $\delta^{18}O$ variations are expected to be closely connected to changes in AMOC due to the
position of the ice core site on the Atlantic sector of the East Antarctic plateau. For other Antarctic
sites, the change of slope around 16.2 ka is less clear, probably due to the damping effect of the
Southern Ocean or because other climatic effects linked to atmospheric teleconnections with the
tropics affect the Pacific and Indian sectors of Antarctica (Stenni et al., 2011, WAIS Divide members,
2013 Buiron et al., 2012). A change in the teleconnections between West Antarctic climate and tropical
regions is also observed around 16.2 ka (Jones et al., 2018). Summarizing, our synthesis of ice core
records clearly demonstrates a climate shift at 16.2 ka, identified in proxy records sensitive to shifts in
tropical hydrology ($CH_4$), mid-latitude hydrological cycle changes in the Atlantic basin (Greenland



second order isotopic tracers), as well as in Antarctic climate dynamics in the Atlantic basin. This
suggests some reorganization of water cycle in the Atlantic region (possibly involving AMOC) related
to surface shifts in the ITCZ at 16.2 ka. This does not appear to affect the high latitudes of the North
Atlantic as Greenland temperatures stay uniformly cold.
At low latitudes, an ITCZ shift at 16.2 ka is clearly expressed through a weak monsoon interval in East
Asian speleothem records and through change in hydrology in the low-latitude Pacific region and Brazil
(Partin et al., 2007; Russell et al., 2014; Strikis et al., 2015). Since we have ruled out a local temperature
signal at 16.2 ka in Greenland, the origin of the Greenland d-excess and $^{17}$O-excess changes around
16.2 ka is also linked to changes in the climate of the source evaporative regions. When evaporation
conditions change, they affect the proportion of kinetic versus equilibrium fractionation, and cause
similar trends in both d-excess and $^{17}$O-excess. Both of them indeed increase when kinetic fractionation
is more important, i.e. when relative humidity decreases, or when a change in sea ice modifies the
evaporative conditions (Klein et al., 2015; Kopec et al., 2016). However, d-excess in the atmospheric
vapor is affected by distillation toward higher latitudes, and strongly depends on the source-site
temperature gradient, while $^{17}$O-excess preserves better the initial fingerprint of relative humidity of
the evaporative region.
As a result, the opposing trends observed in d-excess and $^{17}$O-excess at 16.2 ka can most probably be
explained by an increase of both the relative humidity and the sea surface temperature of the
evaporative source regions for Central and North Greenland. Despite known limitations (Winkler et al.,
2012, Schoenemann and Steig, 2016), the classical approach for inferring changes in source relative
humidity and surface temperature from d-excess and $^{17}$O-excess in Greenland (Masson-Delmotte et
al., 2005a; Landais et al., 2012) suggests respective increases of the order of 3°C and 8% for
temperature and relative humidity of the source evaporative regions respectively. The larger d-excess
increase at the transition between Phase 1 and Phase 2 of Heinrich Stadial 1 observed at GRIP
compared to NGRIP is compatible with a larger proportion of GRIP moisture provided by the mid-



latitude North Atlantic for this site. A larger increase in the sea surface temperature of the source of
moisture for GRIP compared to NGRIP would also reduce the source-site temperature gradient and is
fully compatible with the 2 ‰ less depleted level of $\delta^{18}O$ at GRIP, compared to NGRIP, during Phase 2.
The increases in both temperature and relative humidity of the Greenland source regions suggest a
more intense evaporative flux from lower latitudes starting at 16.2 ka. Such features could be
explained either by a local climate signal of evaporative regions or by a southward shift of evaporative
source regions toward warmer and more humid locations. This latter interpretation is in line with
earlier interpretations of Greenland d-excess changes (Steffensen et al., 2018; Masson-Delmotte et al.,
2005b). The Greenland signals may also be at least partly explained by wetter conditions in the
continental North America evaporative source regions, which are known to partly affect Greenland
moisture today in addition to the main source in Northern Atlantic [38]. This relative humidity signal
reconstructed from Greenland $^{17}O$-excess at the transition between Phase 1 and Phase 2 of Heinrich
Stadial 1 coincides with the onset of the "big wet" period in North American records (Broecker and
Putnam, 2012).
We now explore paleoceanographic records to search for a fingerprint of climate and/or AMOC
reorganization at 16.2 ka in the North Atlantic region and possible implications for our ice core records.
Such comparison of ice core and marine sediment records appears insightful despite existing
limitations attached to relative chronologies. First, high resolution proxy records of surface sea
temperature in the East Atlantic, near Europe, depict a clear warming in the middle of Heinrich Stadial
1 (Bard et al., 2000; Matrat et al., 2014, Figure 3). This signal is coherent with our interpretation of
Greenland d-excess increase at 16.2 ka. In the deep Western Atlantic, no specific feature emerges
between Phase 1 and Phase 2 of Heinrich Stadial 1 from the multi-centennial resolution record of
Pa/Th, a proxy of AMOC strength (McManus et al., 2004). By contrast, a Pa/Th record from the Iberian
margin (Gerhardi et al., 2005) at shallower depth (1500 m shallower than the western Atlantic record)
shows a significant increase at 16.2 ka. These records may be interpreted as follows. A first



modification of the glacial oceanic ventilation occurs at deep depth as early as 18 ka. At 16.2 ka, AMOC
may be further destabilized to additionally affect Pa/Th at shallower depths.
Heinrich Stadial 1 is associated with at least two major Iceberg Rafted Debris (IRD) discharges first
identified near the Iberian margin (Bard et al., 2000). They may reflect either the impact of changes in
ocean conditions on ice shelf and ice sheet stabilities (Alvarez-Solas et al., 2011). Alternatively, the
iceberg discharges themselves may have affected the AMOC, which is known to have major impacts
on patterns of sea surface temperature, sea ice, atmospheric circulation, and climate over surrounding
continents. The first IRD phase originated from ice sheet discharges from Northern Europe and Iceland,
causing strong reorganizations in deep circulation of the North East Atlantic (Stanford et al., 2011,
Grousset et al., 2001; Peck et al., 2006) while the second IRD phase is caused by discharges from the
Laurentide ice sheet. Recent studies (e.g. Hodell et al., 2017, Toucanne et al., 2015) suggest that all IRD
phases occur after 16.2 ka, during Heinrich Stadial 1 Phase 2. Before that, Heinrich Stadial Phase 1 is
associated with a strong increase of sediment fluxes due to meltwater arrival through terrestrial
terminating ice streams originating from both European and American sides of the North Atlantic as a
response to the beginning of the deglaciation (Toucanne et al., 2015, Ullman et al., 2015, Leng et al.,
2018) (Figure 3). During the first slowdown of AMOC during Phase 1 of Heinrich Stadial 1, the
associated warming of subsurface water would hence enable the destabilization of marine ice-shelves
occurring during Phase 2 (Alvarez-Solas et al;, 2011). This second phase of Heinrich Stadial 1 is also
associated with an extensive sea ice production, south of Greenland (Hillaire-Marcel and De Vernal,
2008). The increase of North Atlantic sea ice extent and major iceberg discharges during the second
phase of Heinrich Stadial 1 are coherent with a southward shift of the evaporative region providing
moisture to Greenland supported by d-excess data, and a southward shift of tropical rainbelts (Chiang
and Bitz, 2005), affecting southern hemisphere $CH_4$ sources (Rhodes et al., 2015).
**Conclusions**



Combined measurements of d-excess and $^{17}$O-excess along the NGRIP ice core demonstrate a
decoupling between a cold and stable Greenland climate and changes in hydroclimate at lower
latitudes during the Heinrich Stadial 1, also referred to as the "Mystery Interval" (Denton et al., 2006).
While Greenland temperature remains mostly stable from 20 to 14.7 ka, a large-scale climatic
reorganization takes place at 16.2 ka, associated with warmer and wetter conditions at the location of
Greenland moisture sources. Based on a coherent temporal framework linking the different ice core
records, we show that this event coincides with changes in the characteristics of the bipolar seesaw
pattern as observed in the Atlantic sector of Antarctica, and has a fingerprint in global atmospheric
composition through sharp changes in atmospheric $CO_2$ and $CH_4$ concentrations.
Based on these new ice core records, their coherent chronology, and the comparison with marine and
terrestrial records, we propose the following sequence of events during the last deglaciation. First, the
initiation of Heinrich Stadial 1 occurs at 17.5 ka or earlier, with meltwater arrival from the terrestrial
terminating ice-streams synchronous with a decrease in the North Atlantic sea surface temperature
off-shore Europe, a first AMOC slowdown, drier conditions in North America, and an increase in
Antarctic temperature as well as in atmospheric $CO_2$ and $CH_4$ concentrations. No fingerprint of this first
phase of Heinrich Stadial 1 is identified in Greenland water stable isotope records: $\delta^{18}O$ (and thus local
temperature), $^{17}$O-excess and d-excess remain stable. A possible explanation for such stability is that
the high-latitude warming induced by the increase in the summer insolation at high latitude over the
beginning of the deglaciation is counterbalanced in Greenland by regional changes in e.g. increased
albedo due to sea ice extent or reduced transport of heat by the atmospheric circulation towards
central Greenland, which both can result from a reduced AMOC strength. The global event occurring
at 16.2 ka marks the onset of the second phase of Heinrich Stadial 1. It is associated with (i) strong
iceberg discharges due to dynamical instability of the Laurentide ice sheet, probably induced by the
accumulation of subsurface ocean heat due to a slowdown of AMOC during Phase 1, (ii) a widespread
reorganization of the atmospheric water cycle in the Atlantic region, with significant changes in d-
excess and $^{17}$O-excess in Greenland, as well as (iii) the initiation of weak monsoon interval in East Asia



and (iv) the transition from a "big dry" episode to a "big wet" episodes in North America. We note that
this sequence of events within Heinrich Stadial 1 is invisible in all available Greenland temperature
proxy records, which only display an abrupt warming at the onset of the Bølling-Allerød (14.7 ka).
Attached to a bipolar synchronised chronological framework, our new ice core data provide a unique
benchmark to test the ability of Earth system models to correctly resolve the sub-millennial
mechanisms at play during the last deglaciation, and especially the relationships between meltwater
fluxes, the state of the North Atlantic ocean circulation, the Laurentide ice sheet instability, changes at
the moisture sources of Greenland ice cores, the response of hydroclimate at low and high latitudes,
as well as the net quantitative effects on global methane and carbon budgets.

**Acknowledgements**

The research leading to these results has received funding from the European Research Council
under the European Union's Seventh Framework Programme (FP7/2007-2013) / RC agreement
number 306045. E. C. is funded by the European Union's Seventh Framework Programme for
research and innovation under the Marie Skłodowska-Curie grant agreement no 600207. R.H.R.
received funding from a European Commission Horizon 2020 Marie Sklodowska-Curie Individual
Fellowship (no. 658120, SEADOG).

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

- _d-excess from the NGRIP ice core after correction of the shift between INSTAAR SIL and LSCE (dark green) dataset, and d-excess from the GRIP ice core (light green)._
- _$\delta^{18}O$ from the NGRIP ice core (dark blue) datasets, $\delta^{18}O$ from the GRIP ice core (light blue)._

_Grey intervals display Heinrich Stadials (HS)._



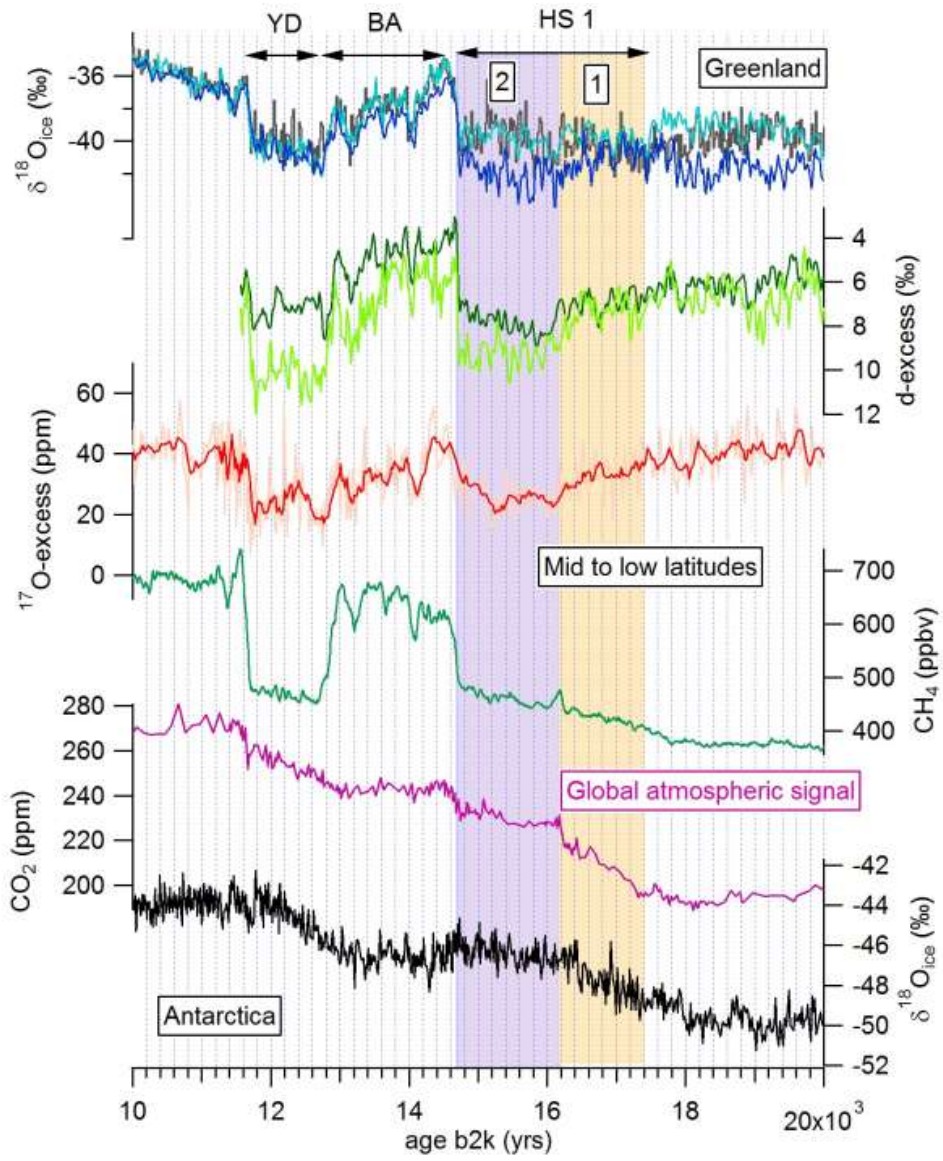


*Figure 2: A synthesis of ice core records over the last deglaciation on the synchronized GICC05/AICC2012*
*timescales with an identification of two phases (1, orange box and 2, purple box) within Heinrich Stadial*
*1 (HS1) as discussed in the text: we locate the transition between phases 1 and 2 at the timing of the*
*sharp increase in CO₂ and CH₄ concentrations, both being global atmospheric composition signals. The*
*Younger Dryas (YD) and Bølling-Allerød (BA) periods are also indicated.*
*From top to bottom:*
*  -  GRIP, NGRIP and GISP2 $\delta^{18}O$ (light blue, dark blue and black respectively (Grootes et al., 1993;*
*     NGRIP community members, 2004) interpolated at a 20 years resolution*
*  -  GRIP and NGRIP d-excess (light and dark green respectively: Jouzel et al., 2005, this study)*
*     interpolated at a 20 years resolution*



-    *NGRIP $^{17}O$-excess (orange curve shows the original series and the red curve the 5 years running*
571          *average, this study)*
-    *WAIS Divide CH$_4$* (Rhodes et al., 2015)
-    *WAIS Divide CO$_2$ (Marcott et al., 2013)*
-    *EPICA Dronning Maud Land (EDML) $\delta^{18}O_{ice}$ (EPICA community members, 2006)*





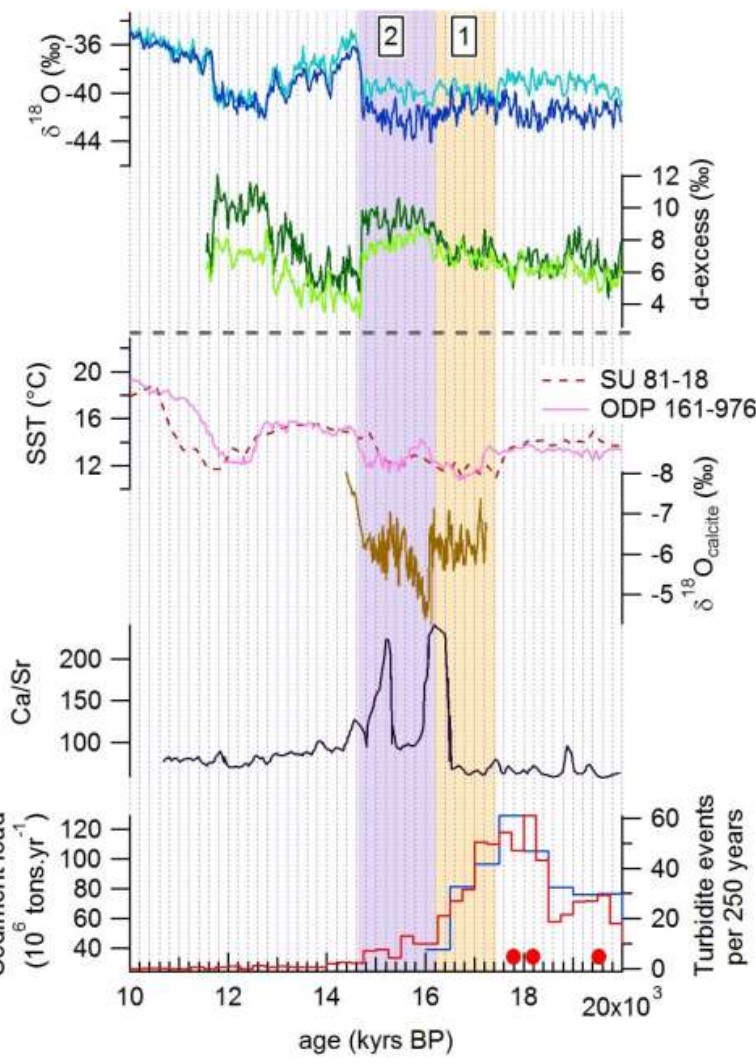


**Figure 3**: *The sequence of Phase 1 and Phase 2 of Heinrich Stadial 1 identified in Greenland records*
*and in proxy records of North Atlantic SST, IRD events, and changes in East Asian hydroclimate. From*
*top to bottom:*

- *NGRIP (dark blue) and GRIP (light blue) $\delta^{18}O$ records*
- *NGRIP (dark green) and GRIP (light green) d-excess records*
- *Sea surface temperature (SST) for North Atlantic cores SU 81-18 (Bard et al., 2000) and ODP*
  *161-976 (Martrat et al., 2014).*
- *Calcite $\delta^{18}O$ of Hulu cave (China, Zhang et al., 2014)*
- *Ca/Sr from site U1308 in the IRD belt (Hodell et al., 2019) as signature from strong iceberg*
  *discharges from the Laurentide ice sheet.*
- *Indications for Channel River sediment load (blue, sediment load; red, turbidite frequency)*
  *(Toucanne et al., 2010; 2015) as signature for meltwater input from European side. The 3 red*
  *circles indicate plumite layers resulting from outburst floods on the Eastern Canadian margin*



*(Leng et al., 2018), i.e. meltwater arrival from the North America side in the absence of strong iceberg discharge.*

*The dashed horizontal line separates the ice core records on the GICC05 timescale from non ice core records on their own timescales.*