# Peer review of "temperature during the last deglaciation Amaëlle Landais1,\*, Emilie Capron2, 3, Valérie Masson-Delmotte1, Samuel Toucanne4, Rachael 3 Rhodes5, Trevor Popp2, Bo Vinther2, Bénédicte Minster1, Frédéric Prié1"

_Climate of the Past, 2018_

## Referee Comment (RC1) · C. Buizert (Referee) · 10 Jul 2018

Landais et al. discuss d-excess and 17O-excess records during Heinrich stadial 1, that provide additional evidence for a shift in atmospheric circulation and the mid-latitude hydrological cycle around 16.2 ka BP, as previously seen in e.g. atmospheric CH4, speleothems, Cariaco basin and North American lake levels. The authors emphasize the point that this event is not seen in Greenland d18O, suggesting a decoupling of Greenland climate from mid-latitude hydrology.

Their data and observations make a valuable contribution to the literature, and I broadly agree with their interpretation. The paper is well-written and easy to follow. I have a

few comments and suggestions that I'd like to see addressed before final publication. After minor revision the paper will be suitable for publication.

Could you speculate on the physical mechanisms for the decoupling? In other words, why are H-events not recorded in Greenland temperature records? My sense has always been that Greenland temperatures are "saturated". Once the winter sea ice edge is far enough south (say 45N), driving it further south by additional AMOC weakening will not cool Greenland any further. Other explanations are also possible of course. Since the decoupling is the main topic of the paper, it would be appropriate to have some discussion on what could cause such a decoupling.

Line 63-64: The 16.2 ka event is also seen very beautifully in Cariaco basin reflectance (Deplazes et al., 2013). Since Cariaco is such an iconic record, it would be worth mentioning this (or even showing it in one of the figures).

Another paper that should be referenced here is (Zhang et al., 2016). They argue for stronger links to Antarctica than to Greenland, as also proposed here. The decoupling between Greenland and mid-latitude hydrology had been noted explicitly by others also, which could be acknowledged more clearly (Rhodes et al., 2015; Zhang et al., 2016; Zhang et al., 2014).

The motivation on line 70-76 for using ice cores is not very strong, in my view. Since there is no H-event signal in Greenland temperature records anyway, there is no benefit in having a hydrological record on the same time scale! All the other records (CH4 and CO2 from Antarctica, sediments and speleothems) are on independent chronologies. Why not just state that ice core dxs and 17O can provide additional evidence to supplement what we know already from sediments and speleothems? That is a strong enough motivation in my view.

Line 81: Please just call it Maximum Counting Error, and leave out the 1 sigma. I know that it is often advertised in terms of standard deviations, but I think it's incorrect. A 200 yr MCE means that GICC05 encountered 400 uncertain layers (each counted as half a

year, representing a 50-50 chance the layer is real). So the 200 yr error is an extreme case where 400 coin flips all landed face-up. Not exactly a 1 sigma event.

Line 86: "using a PICARRO laser cavity..." (change word order)

Line 101: INSTAAR (typo)

Line 156: "reflect" instead of "reflects"

Line 194: "the same timescale". Same timescale as what? As the Greenland records? Did you plot the Rhodes et al. record on its original timescale, or a different one? The caption to Fig. 2 suggests records are on the GICC05/AICC2012 time scale. How did you convert the WAIS Divide records to GICC05?

Line 195: "hypothesized to reflect" instead of "understood to reflect".

Line 199: "...carbon fluxes and/or enhanced air-sea..." Both could be true.

Line 247: A southward shift of source regions is also what I would expect. This could explain the apparent SST increase of the source. However, increasing both RH and SST is hard to do through meridional shifts in atmospheric circulation. SST decreases with latitude, but RH increases. So at lower latitude, RH should be lower, actually. Any thoughts on what circulation change could cause both signatures?

Line 254: how does the "big wet" transition fit in dynamically? Presumably the storm tracks and polar jet stream over N-America shift southward (Asmerom et al., 2010)?

Line 260: The Pa/Th discussion is hard to follow without seeing the data. Please remind the reader that more positive values mean weakened circulation. I am no expert on this proxy, but my understanding is that Pa/Th integrates over the water column via particle scavenging. So I am not sure one can interpret the depth of the site as the depth to which the AMOC was affected. Of course AMOC changes at the surface and at depth are linked. The Pa/Th discussion should be clarified or left out.

Line 283: Ice shelf destabilization by subsurface warming was suggested independently by (Marcott et al., 2011); please cite both.

Line 293: The apparent stability from 20-14.7ka is somewhat misleading, because we know Greenland must have warmed in response to CO2 and insolation. I think this is due to a masking effect; (summer) warming due to insolation and CO2 rise is masked in Greenland temperature records by winter cooling during HS1 driven by AMOC weakening (Buizert et al., 2018). That explains why Greenland and the Laurentide retreat significantly prior to 14.7ka, while it appears there is no warming in Greenland records.

Line 296: the link to EDML had also been suggested by Zhang et al. (2016), and possibly others?

Line 299: consider removing "their coherent chronology". I don't think this adds to much new insight, personally.

Figures: Please add panel labels 1a, 1b, 1c etc, which will make it easier to look up in the caption, and refer to specific records in the text.

Line 585: Should this be Hodell et al. 2017?

Asmerom, Y., Polyak, V.J., Burns, S.J., 2010. Variable winter moisture in the southwestern United States linked to rapid glacial climate shifts. Nature Geosci 3, 114-117.

Buizert, C., Keisling, B.A., Box, J.E., He, F., Carlson, A.E., Sinclair, G., DeConto, R.M., 2018. Greenland-Wide Seasonal Temperatures During the Last Deglaciation. Geophys. Res. Lett. 45, 1905-1914.

Deplazes, G., Luckge, A., Peterson, L.C., Timmermann, A., Hamann, Y., Hughen, K.A., Rohl, U., Laj, C., Cane, M.A., Sigman, D.M., Haug, G.H., 2013. Links between tropical rainfall and North Atlantic climate during the last glacial period. Nature Geosci 6, 213-217.

Marcott, S.A., Clark, P.U., Padman, L., Klinkhammer, G.P., Springer, S.R., Liu, Z., Otto-Bliesner, B.L., Carlson, A.E., Ungerer, A., Padman, J., He, F., Cheng, J., Schmittner,

A., 2011. Ice-shelf collapse from subsurface warming as a trigger for Heinrich events. Proc. Natl. Acad. Sci. U. S. A.

Rhodes, R.H., Brook, E.J., Chiang, J.C.H., Blunier, T., Maselli, O.J., McConnell, J.R., Romanini, D., Severinghaus, J.P., 2015. Enhanced tropical methane production in response to iceberg discharge in the North Atlantic. Science 348, 1016-1019.

Zhang, H., Griffiths, M.L., Huang, J., Cai, Y., Wang, C., Zhang, F., Cheng, H., Ning, Y., Hu, C., Xie, S., 2016. Antarctic link with East Asian summer monsoon variability during the Heinrich Stadial–Bølling interstadial transition. Earth Planet. Sci. Lett. 453, 243-251.

Zhang, W., Wu, J., Wang, Y., Wang, Y., Cheng, H., Kong, X., Duan, F., 2014. A detailed East Asian monsoon history surrounding the 'Mystery Interval'derived from three Chinese speleothem records. Quat. Res. 82, 154-163.

---

## Referee Comment (RC2) · Anonymous Referee #2 · 13 Aug 2018

I think this a great study which is appropriate for the journal. My background is in triple-oxygen from a perspective outside of the hydrology community so I will stick to this. Although the change in 17O-excess is not clearly located at 16.2 ka (as it is really indicated by dD), the interpretation that the trend in 17O-excess over this time period is due to an increase in the relative humidity of the source region through HS1 seems valid. I have no issues with the interpretation and most everything seems to be consistent with previous work. I do have some question on the 17O-excess data itself as mentioned in my most major comment below.

Line 78: The definitions for $\delta$18O, 17O-excess and D-excess should be given at first

[Figure]

occurrence.

My biggest issue with the study is the analytical methods from line 102 to 108 for the CoF3 based water fluorination and 17O calibration. The methods for this are insufficient. The methods as is simply state that the reaction is done and gives a citation for Barkan and Luz 2005. High precision 17O-excess measurements require extreme care in gas handling, the right mass-spec setup and consistent methodology. Some errors in gas handling can generate incorrect values at the 100 ppm level, let alone at the 5 ppm level reported as precision. Impure gas in particular can even yield a false measured relationship between $\delta$′18O and 17O-excess through scale distortion. The minimum things that would need to be known to trust the data in no particular order and not intended to be exhaustive are: 1: What mass-spec is being used? 2: How is the sample being introduced to the mass spec. 3: What is the composition of the in-house reference relative to the samples (raw data would do). 4: Some basics about the CoF3 technique. (He carrier gas?, source of the CoF3 for purity questions, reaction temperature etc.) 5: How is the resulting O2 gas purified? This last point is in particular critical. Sample purification is something that for a long time went overlooked because there was no need to push precision, but to get down to sub-10 ppm with any hope of being similarly accurate seems to require GC purification of the gas. This includes CoF3 lines. If there is no GC purification to remove residual impurities, then I think some in the community would be inclined to not trust the 17O-excess results to the detail needed for the submitted study. These impurities may be things that would clearly cause issues with the measurement, such as the mass 33 isobar generating NF3 but also non-isobar impurities which can generate pressure baseline type effects via scattering. The errors in 17O-excess induced by these impurities roughly scale with the $\delta$18O (or $\delta$17O) so correcting for these can not be done by simply shifting the results but can be accounted for by applying a scale compression correction such as VSMOW-SLAP assuming that the impurity is a constant.

Related to this, how was, or was, the 17O-excess data scaled? Is VSMOW-SLAP

being used? It is stated on line 106 that "home" standards, which I guess should be "in-house" standards, spanning the SMOW-SLAP scale are run on a regular basis, but it is not clear if these have been calibrated to VSMOW-VSLAP2, or any other scaling. It might be this was intended by the Schoenemann et al. 2013 citation, which with the current language seems out of place. The reported values do seem like they plot in the correct region for being calibrated to VSMOW-SLAP.

Line 139-140 This is true enough that seawater 17O-excess remains constant, at least in the recent past. However, this is an artifact of the 0.528 slope in the definition (and the value assigned to SLAP) in addition to the logarithmic form. There should be a citation here that amounts to essentially saying average glacial water falls on a 0.528 slope from modern seawater. Zach Sharp and company had a recent study in Geochemical Perspectives Letters which has data this could be calculated from.

Line 197 "Rhodes 2915" is cited.

As a minor point: In a general sense, I feel there is too much interpretation in the Results section. These instances (e.g. line 185) should be moved to the discussion.
* * *

---

## Author Comment (AC1) · 31 Aug 2018

We thank the two reviewers for their time and valuable comments that were taken into account as detailed in the following.

Comments from Christo Buizert.

- Could you speculate on the physical mechanisms for the decoupling? In other words, why are H-events not recorded in Greenland temperature records? My sense has always been that Greenland temperatures are "saturated". Once the winter sea ice edge is far enough south (say 45N), driving it further south by additional AMOC weakening will not cool Greenland any further. Other explanations are also possible of course. Since the decoupling is the main topic of the paper, it would be appropriate to have some discussion on what could cause such a decoupling.

>> We propose to add the following text: "Uniformly cold conditions are generally observed during Heinrich Stadials of the last glacial period with temperature and $\delta^{18}O$ levels that are not significantly lower than temperature levels observed during Greenland Stadials (Kindler et al., 2014; Guillevic et al., 2014). Because Greenland is surrounded by large sea ice extent during Greenland Stadials and Heinrich Stadials (Hoff et al., 2016), an explanation may be that central Greenland temperatures are saturated during cold periods so that AMOC modifications occurring south of the sea ice edge are not influencing significantly Greenland temperatures. "

- Line 63-64: The 16.2 ka event is also seen very beautifully in Cariaco basin reflectance (Deplazes et al., 2013). Since Cariaco is such an iconic record, it would be worth mentioning this (or even showing it in one of the figures).

>> We propose to add this reference in the following sentence: "At low latitudes, an ITCZ shift at 16.2 ka is clearly expressed through a weak monsoon interval in East Asian speleothem records and through change in hydrology in the low-latitude Pacific region, Cariaco Basin and Brazil (Partin et al., 2007; Deplazes et al., 2013; Russell et al., 2014; Strikis et al., 2015)."

- Another paper that should be referenced here is (Zhang et al., 2016). They argue for stronger links to Antarctica than to Greenland, as also proposed here. The decoupling between Greenland and mid-latitude hydrology had been noted explicitly by others also, which could be acknowledged more clearly (Rhodes et al., 2015; Zhang et al., 2016; Zhang et al., 2014).

>> The reference to the work of Zhang et al. (2016) has been added (see below) and we propose to modify the present conclusion by adding: "These new measurements hence confirm the previous studies of Zhang et al. (2014, 2016) and Rhodes et al. (2015)."

- The motivation on line 70-76 for using ice cores is not very strong, in my view. Since there is no H-event signal in Greenland temperature records anyway, there is no benefit in having a hydrological record on the same time scale! All the other records (CH4 and CO2 from Antarctica, sediments and speleothems) are on independent chronologies. Why not just state that ice core dxs and 17O can provide additional evidence to supplement what we know already from sediments and speleothems? That is a strong enough motivation in my view.

>> What we want to compare is the Greenland temperature record (high latitude) with lower latitude hydrological changes on a common timescale. In this sense, measuring d18O (+ d15N of air), d-excess and 17O-excess on the same ice core has really a sense and it is much better than comparing speleothem or sediment records with a Greenland ice core record. Such common measurements become increasingly important when going back in

time as larger uncertainties become associated to the record chronologies. We would thus like to keep this motivation for our work.

- Line 81: Please just call it Maximum Counting Error, and leave out the 1 sigma. I know that it is often advertised in terms of standard deviations, but I think it's incorrect. A 200 yr MCE means that GICC05 encountered 400 uncertain layers (each counted as half a year, representing a 50-50 chance the layer is real). So the 200 yr error is an extreme case where 400 coin flips all landed face-up. Not exactly a 1 sigma event.

>> Indeed, this was a mistake to call it a 1 sigma uncertainty. It will be corrected.

Line 86: "using a PICARRO laser cavity: : :" (change word order)

>> done

Line 101: INSTAAR (typo)

>> done

Line 156: "reflect" instead of "reflects"

>> done

Line 194: "the same timescale". Same timescale as what? As the Greenland records? Did you plot the Rhodes et al. record on its original timescale, or a different one? The caption to Fig. 2 suggests records are on the GICC05/AICC2012 time scale. How did you convert the WAIS Divide records to GICC05?

>> Actually, we kept the original timescales for the two cores but did a translation since the WAIS timescale (WD2014) is referred to year 1950 and GICC05 to year 2000. So, the conversion is only to refer both timescales to year 2000. We propose to remove "presented on the same timescale" to prevent any confusion and say that the WAIS results are presented on the WP2014 timescale referred to year 2000.

Line 195: "hypothesized to reflect" instead of "understood to reflect".

>> done

Line 199: ": : :carbon fluxes and/or enhanced air-sea: : :" Both could be true.

>> Yes, that was the aim when writing "and/or"

Line 247: A southward shift of source regions is also what I would expect. This could explain the apparent SST increase of the source. However, increasing both RH and SST is hard to do through meridional shifts in atmospheric circulation. SST decreases with latitude, but RH increases. So at lower latitude, RH should be lower, actually. Any thoughts on what circulation change could cause both signatures?

>> A possibility here would be that we have mixed contributions of continental and marine sources for precipitation in Greenland at that period in agreement with the modeling study of Werner et al. (2001). Continental sources are from North America and we have a transition from a big dry to a big wet in that region at 16.2 ka. A contribution from the big wet North America from 16.2 ka would thus explain an increase of relative humidity from that period.

This idea was probably not expressed clearly enough in the present manuscript and we now propose the following text: "The Greenland signal of source humidity increase may at least partly explained by wetter conditions in the continental North America evaporative source regions, which are known to partly affect Greenland moisture today in addition to the main source in Northern Atlantic (Werner et al., 2001; Langen and Vinther, 2009)."

Line 254: how does the "big wet" transition fit in dynamically? Presumably the storm tracks and polar jet stream over N-America shift southward (Asmerom et al., 2010)?

>> This is indeed a possibility, this was added. Thanks.
"This transition to a big wet period can be explained by a southward shift of the storm tracks and polar jet stream over North America shift during this period (Asmerom et al., 2010). "

Line 260: The Pa/Th discussion is hard to follow without seeing the data. Please remind the reader that more positive values mean weakened circulation. I am no expert on this proxy, but my understanding is that Pa/Th integrates over the water column via particle scavenging. So I am not sure one can interpret the depth of the site as the depth to which the AMOC was affected. Of course AMOC changes at the surface and at depth are linked. The Pa/Th discussion should be clarified or left out.

>> The simplest is probably indeed to remove this discussion which is not needed for our conclusion

Line 283: Ice shelf destabilization by subsurface warming was suggested independently by (Marcott et al., 2011); please cite both.

>> This was added.

Line 293: The apparent stability from 20-14.7ka is somewhat misleading, because we know Greenland must have warmed in response to CO2 and insolation. I think this is due to a masking effect; (summer) warming due to insolation and CO2 rise is masked in Greenland temperature records by winter cooling during HS1 driven by AMOC weakening (Buizert et al., 2018). That explains why Greenland and the Laurentide retreat significantly prior to 14.7ka, while it appears there is no warming in Greenland records.

>> A reference was added to this work:
"During Heinrich 1 occurring during the last deglaciation, the story may be more complicated because of the $CO_2$ concentration and insolation increases. In this case, the occurrence of Heinrich Stadial 1 may counteract the increase in Greenland temperature records induced by $CO_2$ and insolation forcing through winter cooling driven by AMOC weakening as suggested by Buizert et al. (2018)."

Line 296: the link to EDML had also been suggested by Zhang et al. (2016), and possibly others?

>> The reference to Zhang et al. (2016) has been added:
"a link between EDML $\delta^{18}O$ record and low latitude signal over Heinrich Stadial 1 has already been suggested by Zhang et al. (2016)."

Line 299: consider removing "their coherent chronology". I don't think this adds to much new insight, personally.

>> Done

Figures: Please add panel labels 1a, 1b, 1c etc, which will make it easier to look up in

the caption, and refer to specific records in the text.

>> This will be done in the revised version.

We thank the two reviewers for their time and valuable comments that were taken into account as detailed in the following.

---

## Author Comment (AC2) · 31 Aug 2018

We thank the two reviewers for their time and valuable comments that were taken into account as detailed in the following.

Response to Referee 2:

I think this a great study which is appropriate for the journal. My background is in triple-oxygen from a perspective outside of the hydrology community so I will stick to this. Although the change in 17O-excess is not clearly located at 16.2 ka (as it is really indicated by dD), the interpretation that the trend in 17O-excess over this time period is due to an increase in the relative humidity of the source region through HS1 seems valid. I have no issues with the interpretation and most everything seems to be consistent with previous work. I do have some question on the 17O-excess data itself as mentioned in my most major comment below.

Line 78: The definitions for _18O, 17O-excess and D-excess should be given at first occurrence.

>> Done

My biggest issue with the study is the analytical methods from line 102 to 108 for the CoF3 based water fluorination and 17O calibration. The methods for this are insufficient. The methods as is simply state that the reaction is done and gives a citation for Barkan and Luz 2005. High precision 17O-excess measurements require extreme care in gas handling, the right mass-spec setup and consistent methodology. Some errors in gas handling can generate incorrect values at the 100 ppm level, let alone at the 5 ppm level reported as precision. Impure gas in particular can even yield a false measured relationship between _âAˇ š18O and 17O-excess through scale distortion.

>> The method for water $^{17}$O-excess measurements through fluorination was already described in several papers (all details are given in Barkan and Luz, 2005) but we agree with the reviewer that it is important to detail again here the methodology, especially since we improved precision compared to previous studies at LSCE using a new mass spectrometer (MAT 253).

The minimum things that would need to be known to trust the data in no particular order and not intended to be exhaustive are:

1: What mass-spec is being used?

>> MAT 253

2: How is the sample being introduced to the mass spec.

>> To reach the 5 ppm precision, we use the mass spectrometer in dual inlet mode and hence introduce the sample in gas phase (pure oxygen) through the classical sample bellow of the mass spectrometer. The sample is run (measured) against a standard (pure commercial oxygen).

3: What is the composition of the in-house reference relative to the samples (raw data would do) ?

>> The in-house references are several water standards calibrated every 3 years with respect to SMOW and SLAP provided by IAEA. For this study, we used in-house standards

with the following isotopic composition: NEEM ($\delta^{18}O$ = -33.56 ‰ ; $^{17}O$-excess = 32 ppm); OC ($\delta^{18}O$ = -54.05 ‰ ; $^{17}O$-excess = 12 ppm); ROSS ($\delta^{18}O$ = -18.64 ‰ ; $^{17}O$-excess = 37 ppm)

**4: Some basics about the CoF3 technique. (He carrier gas?, source of the CoF3 for purity questions, reaction temperature etc.)**

>> We used the published procedure: He carrier gas is purified through a trap of liquid nitrogen (-196°C); CoF3 is bought by Sigma-Aldrich following numerous tests by several producers (Sigma-Aldrich CoF3 gives the best reproducibility and precision according to the species measured by the mass spectrometer); the temperature reaction is 370°C.

**5: How is the resulting $O_2$ gas purified?**

>> The purification is done through a molecular sieve trap immersed in liquid nitrogen. Tests of GC purification were also performed during the development of the line at LSCE but did not improve the precision. Indeed, a systematic correction to a V-SMOW – SLAP scale is performed.
Every two-three weeks, three in-house standards bracketing the d18O and d17O values of the samples are run on the fluorination line. These standards are then used to put the d18O and d17O values on the V-SMOW – SLAP scale following procedure described in Schoenemann et al. (2013) and Landais et al. (2013). Then, everyday, only one in-house standard is run to check the day to day stability of the whole system (line + mass spectrometer) but this house standard is not used alone for shifting the d17O and d18O data, a full scale compression on the V-SMOW – SLAP scale is performed.
This will be explained better in the new manuscript and we propose the following paragraphs:

"In order to perform $^{17}O$-excess measurements on water samples at LSCE, we follow the method described in details in (Barkan and Luz, 2005). In short, for each sample, 2 mL of water are injected in a helium flow purified by passing through a trap immersed in liquid nitrogen. Water vapor then reacts with CoF$_3$ (producer Sigma-Aldrich) in a nickel tube heated at 370°C to produce oxygen and fluorhydric acid which is trapped in liquid nitrogen at the outlet of the nickel tube. Oxygen is first trapped in a molecular sieve tube immersed in liquid nitrogen and then separated from helium and purified through 2 cycles of warming (+30°C) and cooling (-196°C) of the tube with molecular sieves. The oxygen is finally trapped in a manifold immersed in liquid helium. After warming the manifold at least 40 minutes at room temperature, the triple isotopic composition of produced oxygen is injected in the mass spectrometer (MAT 253) and measured by dual inlet against a reference $O_2$ gas (2 runs of 20 measurements).

Every day, at least one home standard is run with the batch of samples to check the stability of the fluorination line and mass spectrometer and a series of 3 water home standards, whose $\delta^{18}O$ and $\delta^{17}O$ values are calibrated on the SMOW – SLAP scale following Schoenemann et al. (2013), is run at least every month. For this study, the SMOW – SLAP calibrated home standards have $\delta^{18}O$ values of respectively -18.64 ‰, -33.56 ‰ and -54.05 ‰, hence bracketing the $\delta^{18}O$ values of the measured samples. The comparison of the measured and SMOW-SLAP calibrated $\delta^{18}O$ and $\delta^{17}O$ values then enable calibrating the $\delta^{18}O$ and $^{17}O$-excess values of the NGRIP samples of this study following the method described in (Schoenemann et al., 2013; Landais et al., 2014). The resulting mean uncertainty is of 5 ppm (1 $\sigma$) for the $^{17}O$-excess measurements of this study and we note that the use of the MAT 253 mass spectrometer gave much more stable results that a Delta V+ instrument used for previous studies at LSCE (e.g. Landais et al., 2012)."

This last point is in particular critical. Sample purification is something that for a long time went overlooked because there was no need to push precision, but to get down to sub-10 ppm with any hope of being similarly accurate seems to require GC purification of the gas. This includes CoF3 lines. If there is no GC purification to remove residual impurities, then I

think some in the community would be inclined to not trust the 17O-excess results to the detail needed for the submitted study. These impurities may be things that would clearly cause issues with the measurement, such as the mass 33 isobar generating NF3 but also non-isobar impurities which can generate pressure baseline type effects via scattering. The errors in 17O-excess induced by these impurities roughly scale with the _18O (or _17O) so correcting for these can not be done by simply shifting the results but can be accounted for by applying a scale compression correction such as VSMOW-SLAP assuming that the impurity is a constant.

Related to this, how was, or was, the 17O-excess data scaled? Is VSMOW-SLAP being used? It is stated on line 106 that "home" standards, which I guess should be "in-house" standards, spanning the SMOW-SLAP scale are run on a regular basis, but it is not clear if these have been calibrated to VSMOW-VSLAP2, or any other scaling. It might be this was intended by the Schoenemann et al. 2013 citation, which with the current language seems out of place. The reported values do seem like they plot in the correct region for being calibrated to VSMOW-SLAP.

>> See answer above.

Line 139-140 This is true enough that seawater 17O-excess remains constant, at least in the recent past. However, this is an artifact of the 0.528 slope in the definition (and the value assigned to SLAP) in addition to the logarithmic form. There should be a citation here that amounts to essentially saying average glacial water falls on a 0.528 slope from modern seawater. Zach Sharp and company had a recent study in Geochemical Perspectives Letters which has data this could be calculated from.

>> We actually cannot claim for sure that 17O-excess of seawater remains constant over the last deglaciation, no data can demonstrate it with the sufficient accuracy yet. What we want to emphasize is that the water cycle processes will not create an artificial 17O-excess signal linked to the seawater d18O change of 1 permil over the last deglaciation as it is the case for d-excess. If $^{17}$O-excess of seawater is modified, this modification will be conserved in the meteoric water $^{17}$O-excess. This was indeed not very clear and we propose to clarify as follows:

"Because of its logarithmic definition, $^{17}$O-excess is not sensitive to changes in $\delta^{18}O_{\text{sea water}}$ given that the $^{17}$O-excess of global sea water remains constant with time. As a consequence, a change in sea water isotopic composition will only be transmitted to the $^{17}$O-excess of the precipitation if the $^{17}$O-excess of the evaporated sea-water is modified."

Line 197 "Rhodes 2915" is cited.

>> Oups… Indeed, i is not correct and the "9" should be changed in "0", thank you for pointing it.

As a minor point: In a general sense, I feel there is too much interpretation in the Result part:

>> We will try to better equilibrate this part and it goes along with more details in the methodology part answering the comment of reviewer 2 above.